# Variability of Low-Z Inhomogeneity Correction in IMRT/SBRT: A Multi-Institutional Collaborative Study

**DOI:** 10.3390/jcm12030906

**Published:** 2023-01-23

**Authors:** Poonam Yadav, Colleen M. DesRosiers, Raj K. Mitra, Shiv P. Srivastava, Indra J. Das

**Affiliations:** 1Department of Radiation Oncology, Northwest Memorial Hospital, Northwestern University Feinberg School of Medicine, Chicago, IL 60611, USA; 2Department of Radiation Oncology, Indiana University Health, Indianapolis, IN 46202, USA; 3Department of Radiation Oncology, Ochsner Health System, New Orleans, LA 70121, USA; 4Department of Radiation Oncology, Dignity Health System, Phoenix, AZ 85013, USA

**Keywords:** lung cancer, IMRT, heterogeneity correction, dose calculation, algorithm

## Abstract

Dose-calculation algorithms are critical for radiation treatment outcomes that vary among treatment planning systems (TPS). Modern algorithms use sophisticated radiation transport calculation with detailed three-dimensional beam modeling to provide accurate doses, especially in heterogeneous medium and small fields used in IMRT/SBRT. While the dosimetric accuracy in heterogeneous mediums (lung) is qualitatively known, the accuracy is unknown. The aim of this work is to analyze the calculated dose in lung patients and compare the validity of dose-calculation algorithms by measurements in a low-Z phantom for two main classes of algorithms: type A (pencil beam) and type B (collapse cone). The CT scans with volumes (target and organs at risk, OARs) of a lung patient and a phantom build to replicate the human lung data were sent to nine institutions for planning. Doses at different depths and field sizes were measured in the phantom with and without inhomogeneity correction across multiple institutions to understand the impact of clinically used dose algorithms. Wide dosimetric variations were observed in target and OAR coverage in patient plans. The correction factor for collapsed cone algorithms was less than pencil beam algorithms in the small fields used in SBRT. The pencil beam showed ≈70% variations between measured and calculated correction factors for various field sizes and depths. For large field sizes the trends of both types of algorithms were similar. The differences in measured versus calculated dose for type-B algorithms were within ±10%. Significant variations in the target and OARs were observed among various TPS. The results suggest that the pencil beam algorithm does not provide an accurate dose and should not be considered with small fields (IMRT/SBRT). Type-B collapsed-cone algorithms provide better agreement with measurements, but still vary among various systems.

## 1. Introduction

Lung cancer is one of the most common malignancies in both incidence and mortality, and is a leading cause of cancer-related deaths worldwide [1]. Irrespective of the stage of lung cancer, radiotherapy techniques such as three-dimensional conformal radiotherapy (3DCRT) and intensity-modulated radiotherapy (IMRT/VMAT) are considered effective treatment techniques with a high level of control over local progression [2,3]. In modern times, more cases are treated with stereotactic body radiation therapy (SBRT). The improved treatment targeting of contemporary radiation therapy modalities and improved accuracy of dose-calculation algorithms are critical to the success of these procedures, as shown in clinical trials [4,5]. Additionally, dose prescription and recording should be in accordance with international guidelines such as ICRU-83 and ICRU-91 [6,7] to allow outcome comparisons.

Radiation therapy with concurrent chemotherapy is an established treatment regimen for locally advanced non-small cell lung cancer [8,9,10]. Typically, 3DCRT for lung cancer is commonly used by multiple clinics to achieve better conformity of the target volume but simultaneously increases the volume of normal tissue receiving high doses [8,11,12]. In contrast to this, IMRT/VMAT considerably reduces the dose to the surrounding organs at risk (OARs) based on constraints that produce high-gradient desired dose distribution [7,8,13,14,15].

Radiotherapy has been successfully employed as a primary treatment in different parts of the body with varying densities based on water, soft tissue, air, and bones. For some conditions, such as high-density implants, there are larger studies investigating discrepancies in dose calculations [16,17,18,19,20,21,22]. Shiraishi et al. [23] used a lung phantom and concluded that analytical anisotropic algorithms (AAA) and Acuros-XB provided an accurate dose compared with the measurements, but did not compare the variability from different types of algorithms that differ among vendors in small-fields. Various other studies indicate the variability of the inhomogeneity correction in a select class of algorithms [24,25,26,27]. Comparing older algorithms (equivalent path length and pencil beam) with Monte Carlo, it was noticed that pencil beam provided a 20–30% higher dose compared with Monte Carlo and correlated its impact on SBRT tumor control probability [25,27]. Nevertheless, it is apparent that further studies are required to see the impact of different dose-calculation algorithms in low-density mediums.

A recent survey conducted by the authors’ group (see in Section 2) found that, routinely, nearly 1/3 of American radiation oncology centers do not use inhomogeneity corrections. When assessed in terms of 3DCRT and IMRT, 83% of centers were found to use inhomogeneity correction in IMRT/VMAT. These practices persist despite several high-profile publications advocating inhomogeneity correction. Orton et al. [28], even in 1984, advocated that inhomogeneity correction should be used. Lung cancer studies with and without tissue-density corrections have shown wide variation in radiation dose calculations. Various clinical protocols did not recommend inhomogeneity correction until Xiao et al. [29], in 2011, showed the dosimetric impact of the inhomogeneity correction where isodose volume for the prescribed dose may decrease with inhomogeneity corrections [29]. This led the Radiation Therapy Oncology Group (RTOG) to modify the prescription dose.

It is a well-established fact that SBRT procedures deliver fairly high radiation doses to smaller volumes and hence precision in dosimetry is extremely important. Therefore, the efficacy of inhomogeneity corrections is of paramount importance. The AAPM Task Group 85 guidelines for lung suggested that peripheral doses of tumors should be calculated precisely with the consideration of the loss of electron equilibrium and dose reduction along the central axis and near the beam edge [30]. Thus, inhomogeneity corrections should be considered for dose calculation at the interface of tumor and lung [19]. Inhomogeneity corrections have evolved over 60 years, with the generation of new algorithms offering incrementally better accuracy. Knöös et al. [18] described the dose-calculation algorithms broadly in two classes. Type A includes older algorithms (Clarkson, Batho, Power law, equivalent path length, etc.), pencil beam, and convolution with an invariant kernel. Type B includes Collapsed Cone Superposition and Linear Boltzmann (e.g., Accuros). Even though Eclipse AAA is a modified pencil beam algorithm, it is included in type B. Additionally, there exists a third type, type C, which includes Monte Carlo implementation in many commercial treatment planning systems (TPS) such as Monaco and RayStation.

To validate the TPS findings a lung phantom was created with cork and solid water for measurements. Special attention was given to measuring dose at depths in phantom with various field sizes, as IMRT/SBRT fields are typically small. This study constitutes a clinical case using phantom measurements for lung cancer with inhomogeneity corrections in low-density (lung cancer) with different dose-calculation algorithms.

## 2. Materials and Methods

As described above, a survey was performed to gauge practices of lung radiation treatment in the USA. The survey was randomly distributed to 200 institutions (based on our acquaintances, but mainly academic). The questionnaire included queries about treatment machine, inhomogeneity correction, technique (3DCRT, IMRT, etc.), TPS, and version and dose-calculation algorithms. Only 68 of the contacted institutions responded.

The CT data of an anonymized lung cancer patient with Institutional Review Board (IRB) exemption and target volume and OARs was selected for plan comparison. An in-house phantom was built with dimensions of 30 × 30 cm^2^ and 19.4 cm thickness (Figure 1). To represent the lung, pressed sliced cork of variable thickness with a density of 0.25 g/cm^3^ was used. The cork thickness of 14.4 cm was sandwiched between 2.5 cm and 3.5 cm-thick solid water slabs. Figure 1 shows the cork phantom geometry. On the bottom slab, specially designed strips and the chamber holder are shown to measure the dose by changing the order of the sheet. If needed, this phantom can also be used for off-axis measurement. The cork density was chosen to represent in the middle of the human lung densities (0.1 g/cm^3^–0.3 g/cm^3^). The patient and phantom were scanned on a 16-slice Philips Brilliance CT scanner. The scanned CT data with PTV and OAR were sent to nine participating collaborators having different TPS. The dose–volume constraints from primary author’s institution as shown in Table 1 were also sent for IMRT planning for the lung patient. Since different machines are used only 6 MV photon beams were used where depth-dose variation was within ±0.5% among machines of different manufacturers. Monte Carlo [31,32] calculations were performed on the phantom to compare with the other algorithms.

As in-vivo dosimetry in patients requiring lung treatment specifically is not possible, it was performed in a lung phantom, using different algorithms. Measurements were performed in phantom using a single beam with the following square field sizes: 1, 2, 3, 4, 5, 6, 8 and 10 cm^2^. Inhomogeneity correction (IC) or correction factor was calculated as the ratio of D_i_ to D_h_ where D_i_ and D_h_ are doses at the same depth with and without inhomogeneity, respectively, e.g., IC = D_i_/D_h_ in a given field size. Dose measurements were also performed with a 0.125 cm^3^ ion chamber in cork and solid water phantom at various depths (specially drilled hole in cork) and field sizes. The field size study was critical since small fields are used in IMRT/VMAT and it is known that dosimetry is challenging because of loss of lateral equilibrium in small fields [33,34]. The doses calculated with various algorithms were analyzed with respect to the measured dose. Details of the algorithms, vendors, and treatment machines are summarized in Table 2. As described, two classes of algorithms are used in clinical settings, i.e., measurement-based (Clarkson, Batho, equivalent path length, pencil beam; type-A algorithm) and analytical (convolution, and collapsed cone (CC); type-B algorithms). In modern times, most TPS use analytical methods based on the photon kernel derived from Monte Carlo. Hence, pencil beam and CC are two important types to compare, as was performed in this work.

## 3. Results

Our survey also found that various TPS using varied algorithms are used for lung cancer. In this survey, Clarkson/Batho, pencil beam, convolution, and collapsed cone algorithms were used in 4.4%, 26.5%, 16.2%, and 52.9%, respectively. To extend the findings from our survey, a multi-institutional study was conducted to examine lung cancer treatment plans with differing TPS. The lung cancer patients’ dose distributions with identical dose–volume constraints depicted by symbols are shown in Figure 2 from various TPS. The dose–volume histograms (DVH) for target and OARs are known to be extremely variable because of the cost function associated with optimization routines used in various TPS [35]. However, for identical cost functions, the DVH variability seen in Figure 2 constitutes an alarming finding, indicating that multiple factors play a role in the management of the planning, including time factor [36], optimization [35,37], and dose calculation [38]. However, it is sensible to conclude that dose-calculation algorithms play the paramount role in the dose distribution, as shown in Figure 2.

As noted in Table 2, two predominant classes of algorithms are used in clinics, and plans from TPS are summarized in Figure 3 and Figure 4. Figure 3 and Figure 4 show the doses calculated with different field sizes in a lung phantom using pencil beam and CC algorithms, respectively. Note the contrasting shape of the curves in Figure 3 and Figure 4. For pencil beams the curves increased linearly with depth and field size contrary to the CC-type algorithms, where the dose decreased.

The correction factor for pencil beam algorithms increased at all depths. Until the depth of 4 cm, the difference in D_i_/D_h_ was unnoticeable, while with increased depths, it increased steadily. The lower field sizes of 1 × 1 cm^2^, 2 × 2 cm^2^, and 3 × 3 cm^2^ showed slightly higher correction factors compared with larger field sizes, probably due to lateral electron transport that the pencil beam does not handle accurately. The highest correction factor of 1.65 was observed for 1 × 1 cm^2^ field size for the pencil beam algorithms.

For analytical dose-calculation algorithms, the correction factor is around 1.00 for depths up to 3 cm. For lower field sizes, i.e., 1 × 1 cm^2^, 2 × 2 cm^2^, and 3 × 3 cm^2^, the correction factor gradually fell below 1.00, with the lowest values of approximately 0.60 for 1 × 1 cm^2^ at 8 cm depth. At the depth of 10 cm, the correction factor steadily increased for all field sizes and at the depth of 17 cm all correction factors converged around a value of 1.60. A similar pattern was observed for the correction factor in the case of CC algorithms with a negligible difference at the depth of 3 cm and then significant decreases for lower field sizes.

Correction factor results for different field sizes as a function of depth for the selected algorithms from the same TPS (Eclipse) are shown in Figure 4 with different available algorithms: pencil beam, analytical anisotropic algorithm (AAA), and Acuros compared with CC. In the case of Acuros, there was a steep fall in the correction factor between the depths of 3 and 4 cm, and then a steady rise was observed for all field sizes and depths up to 17 cm of the lung tissue interface. Correction factors for all field sizes converged at ~1.5 for a depth of 17 cm and then started to diverge again with increasing depth. These curves (Figure 5) are strikingly different in shape and magnitude. This has crucial clinical implication for patient treatment, which was alluded to by Chetty et al. [27,39]. Additionally, the dosimetric differences are also noted in clinical trials where the prescribed dose was recommended to be modified by 10.1% [29]. It is imperative that, in all clinical trials, quality assurance on dose calculation be verified by various algorithms and should not be blindly accepted. Additionally, as shown in this study, pencil beam should not be used in lung cancer treatment, as recommended by the TG-155 [34]. This is slowly being adopted in most clinical trials from different organizations.

Differences between the measured and calculated doses for different algorithms are shown in Figure 6. Pencil beam algorithms showed significant differences and unusual results. For small fields (1 × 1 cm^2^), 50–70% differences were observed, which reduced to 15–30% for 3 × 3 cm^2^ fields among different TPS. For CC-type algorithms the difference between measured and calculated doses was ±5%, which increased to ±10% for small fields (≤3 × 3 cm^2^). For large field sizes the trend of both categories of algorithms was similar.

## 4. Discussion

It is a well-established fact that SBRT procedures deliver fairly high radiation doses to smaller fields with high dose gradients. Hence, precision in delivery is crucial. Therefore, the efficacy of inhomogeneity corrections is of paramount importance. It is challenging to conduct measurements and dose validation on small fields mainly because of high radiation doses with steep gradients and various physical factors as described in TG-155 [34]. Dose inhomogeneity may also be a contributing factor due to dimensions for the detector with respect to small field size. Jones et al. [40,41] and Carrasco et al. [24] reported that dosimetry significantly varies with different TPS in small fields and in low density mediums. Overall, the success of treatment greatly depends on the accuracy of the dose-calculation algorithms implemented in TPS. This is particularly pertinent since radiation dose calculations for lung SBRT depend on low-density inhomogeneity correction algorithms implemented in TPS. Our results across multiple institutions using different TPS show that it is important to check the accuracy of TPS’ dose-calculation algorithms while considering low-density inhomogeneity corrections.

There are several challenges in lung dosimetry from the perspective of inhomogeneity-corrected treatment planning and delivery. The procedure of assigning unit density to low-density materials per RTOG 0236 protocol is not the best option for accuracy in dose calculation. Inhomogeneity-correction algorithms implemented in TPS and manually applying a differential correction factor can help to optimize the treatment plan. Pencil beam and CC are two widely used inhomogeneity-correction algorithms (26% and 53%, respectively, from our survey data). The pencil beam algorithm considers only the path length in the forward direction and, hence, results in inaccurate dose calculations for small fields, which is generally known.

On the contrary, CC algorithms account for electron transport and outperform pencil beam algorithms with inhomogeneities, especially in small fields. AAPM TG-155 highlighted several limitations of the pencil beam category of algorithms for small fields and provided the guidelines suggesting that pencil beam algorithms should not be relied upon in inhomogeneous mediums. The report suggests that TPS dose calculation based on non-pencil beam algorithms should be utilized for small fields in heterogeneous media. This is supported by our data, as shown in Figure 6, where significant differences (≤70%) between TPS’ calculated and measured doses for pencil beam were observed. The magnitude of dose difference in small fields could vary with a suitable detector such as a plastic scintillator [42] or microsilicon [43]. However, this study did not validate the measured dose, but instead visible patterns with different algorithms.

Pencil beam algorithms are mainly designed around equivalent path length for inhomogeneity corrections and electron transport is not specifically modeled. In addition, these algorithms also sample the density variations in one dimension, i.e., the direction of the primary beam. The algorithms implementing electron transport as well as secondary photon transport with consideration of varying density in three dimensions are complex but add precision in modeling. Such algorithms are likely to present higher agreement with Monte Carlo results. The secondary electron transport and subsequent charged particle equilibrium have a significant impact on dose calculations for small fields within an inhomogeneous medium.

The Boltzmann transport equation used in the Eclipse Acuros algorithm describes the interaction of radiation particles within matter. Acuros comprises two main components: the photon beam source model and the radiation transport model based on spatial, energy, and angular parameters [21]. Studies have shown that Acuros results, both in water and other mediums, agree with Monte Carlo results. Clinical results for non-small cell lung cancer dose calculations with Eclipse implementation of Acuros have shown relatively good consistency for heterogeneous mediums [44]. 

This study shows that the correction factor for the AAA and CC are similar with the smallest correction factor for Acuros. The largest difference was noted in pencil beam algorithms, reflecting the results of the study performed by Ottosson et al. [45] for the selection of beam energy and motion management in lung cancer. These results clearly show the challenges in low-density heterogeneity corrections in pencil beam algorithms which may adversely affect the treatment. Awareness efforts by different vendors, such as warning labels and proper documentation in training materials and manuals, may help in mitigating the discrepancies, especially in low- and middle-income countries and remote centers. Contemporary TPS utilizes a set of algorithms that help in improving the computation time or precision of dose calculations. It is very likely that a typical dose-calculation algorithm, with quicker dose calculation, may not be a viable option considering the inhomogeneities in the thorax region. In high-dose-regimen procedures such as SBRT, it is imperative to have a ≤5% difference in measured versus calculated dose (see algorithms presented in Figure 6).

This study was originally designed for both 6 MV and 15 MV photon beam energies. However, considering the benefits of lower beam energies for low-density mediums, the methodology was revised to investigate only 6 MV. It is a well-established fact that higher beam energies of (15 or 18 MV) have superior depth dose and improved dose uniformity in homogeneous tissue equivalent medium. However, in low-density regions such as lung, high-energy electrons result in electronic disequilibrium in lateral direction and compromise the lesion coverage.

The pencil beam algorithms showed large discrepancies between measured and calculated doses for lower field sizes (~70%). Since beam energy is inversely proportional to the field sizes, for smaller field sizes the beam profile does not plateau. Several studies have highlighted the issue of lateral beam disequilibrium for lung [34]. In other words, TPS fails to accommodate lateral lung inhomogeneity and subsequently the dose is overestimated for low-density regions.

IMRT/VMAT is utilized by a much higher number of institutes compared with 3DCRT for lung radiotherapy, mainly because of the use of SBRT. IMRT has the potential to reduce the volume of low doses for normal tissues and, therefore, reduce the likelihood of radiotoxicity while increasing the integral dose. Hence, this increases the efficacy of the treatment. Technological advancements over time have also led to the popularity of IMRT in treating low-density regions mainly due to improved monitor unit efficiency and good agreement between optimized and delivered intensity maps. Additionally, the standard capabilities of IMRT to modulate the intensity of individual beams play a key role in its widespread popularity.

Clinical studies show that small decreases in radiation dose, as a result of discrepancies in TPS dose-calculation algorithms, can adversely affect local tumor control. This highlights the fact that tissue inhomogeneity should be properly assessed for uniform dose distribution and accuracy in dose calculation. Some limitations of this study are the selection of institutions that have advanced algorithms such as Elekta Monaco and RayStation. Nonetheless, this study shows that pencil beam should not be used for low-density mediums as it does not provide electron transport in the lateral direction [46]. Mostly type-B algorithms, collapsed cone-type as well as Boltzman transport (Acuros XB) are shown to be closer to Monte Carlo calculation and should be used [26].

## 5. Conclusions

Dose-calculation algorithms significantly affect treatment planning and, hence, possibly treatment outcomes. Therefore, discrepancies in calculated doses should be carefully analyzed, especially in the heterogenous regions where dose calculations by majority of the algorithms are likely to deviate from the measured dose. The limitations and the suitability of the TPS and the underlying algorithms should be thoroughly understood for routine clinical use but more importantly in small fields as used in SBRT. It is highly recommended that type A (i.e., pencil beam algorithms) should not be used in modern TPS for lung cancer treatment. Rather, type-B algorithms should be used.

## Figures and Tables

**Figure 1 jcm-12-00906-f001:**
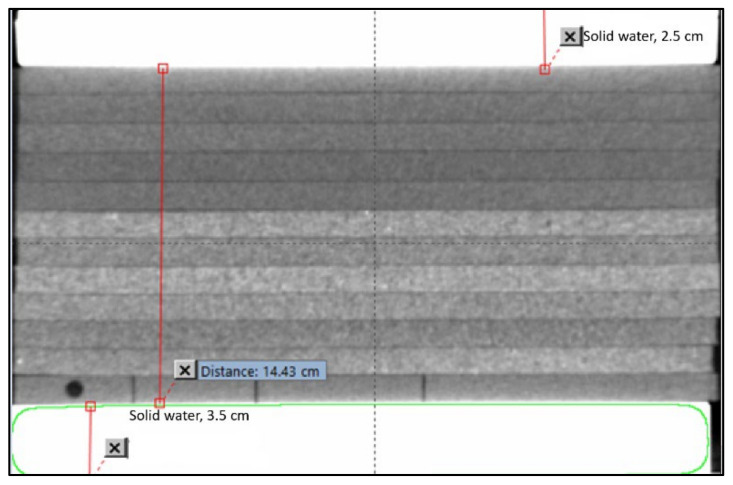
Lung phantom geometry for dose measurement. Slabs of pressed cork with densities of 0.25 g/cm^3^ are used. The bottom cork sheet shows the detector holder and strips of cork that can be rearranged in depths and in the off-axis position.

**Figure 2 jcm-12-00906-f002:**
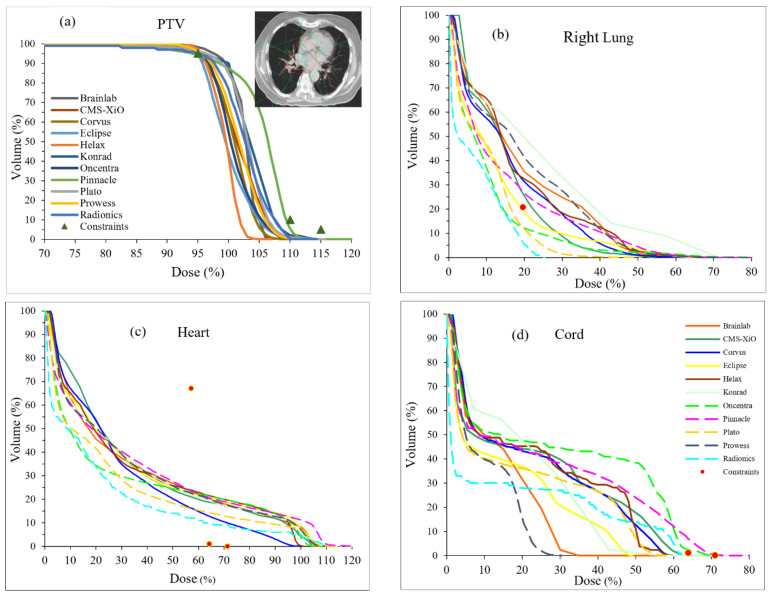
(**a**) Dose–volume histogram of a lung patient from various treatment planning systems. The inset in (**a**) shows the axial CT data of the patient which were sent to every institution. (**b**) Rt lung, (**c**) heart, and (**d**) spinal cord. Please note the variability in dose calculation among eleven systems. Please note that the symbols represent the dose–volume constraints sent to institutions and are required to meet.

**Figure 3 jcm-12-00906-f003:**
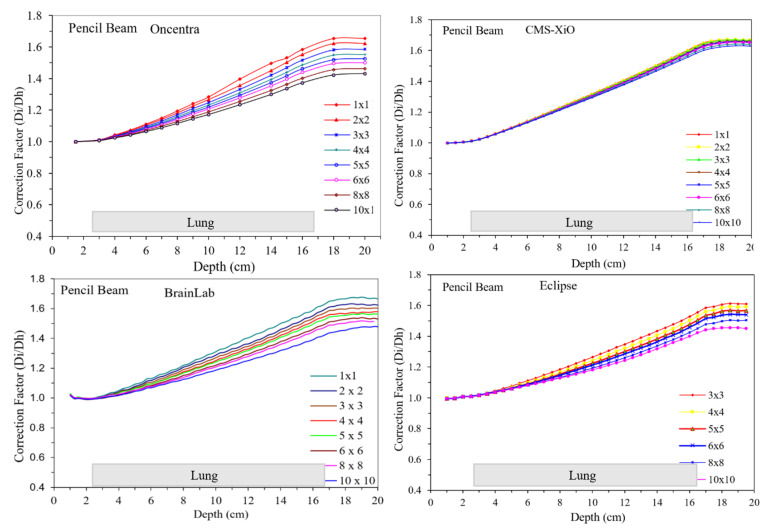
Calculated doses with different field sizes in a lung phantom using pencil beam-type algorithms. Note the location of the lung material and the shape of the various curves. Though trends appear similar, the magnitudes are different.

**Figure 4 jcm-12-00906-f004:**
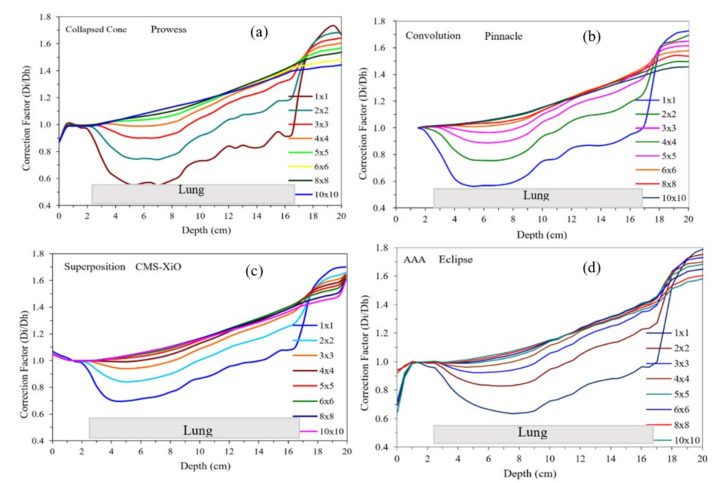
Calculated doses with different field sizes in a lung phantom using various - dose-calculation algorithms; (**a**) Collapsed Cone from Prowess, (**b**) Convolution from Pinnacle, (**c**) Superposition from CMS-XiO and (**d**) AAA from Eclipse. Please note the shape, pattern, and magnitude compared to Figure 3 for the pencil beam.

**Figure 5 jcm-12-00906-f005:**
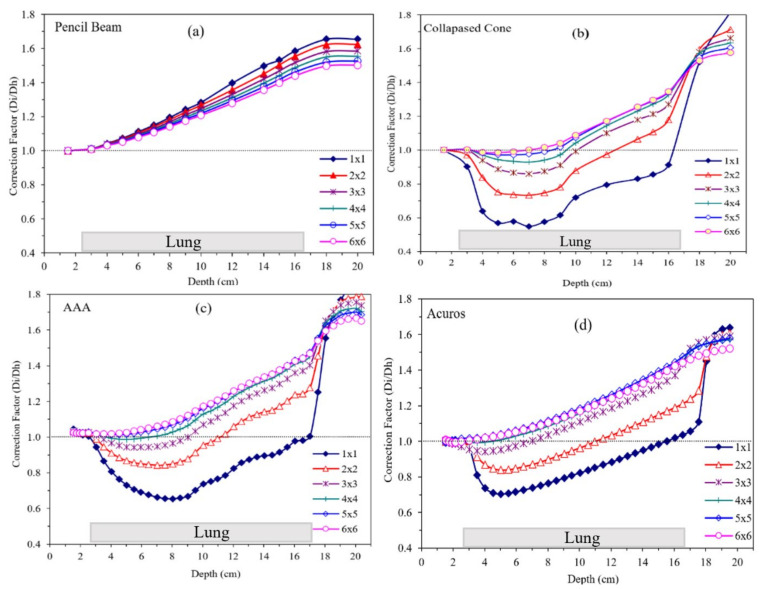
Correction factors for different field sizes as a function of depth for a select (**a**) Pencil Beam, (**b**) Collapsed Cone (**c**), Analytical Anisotropic Algorithm, and (**d**) Acuros XB algorithm. Figure 4a,c,d are from the Varian Eclipse treatment planning system whereas Figure 4b is from Prowess TPS. Please note the stark differences in the shape of the curves with varying magnitudes.

**Figure 6 jcm-12-00906-f006:**
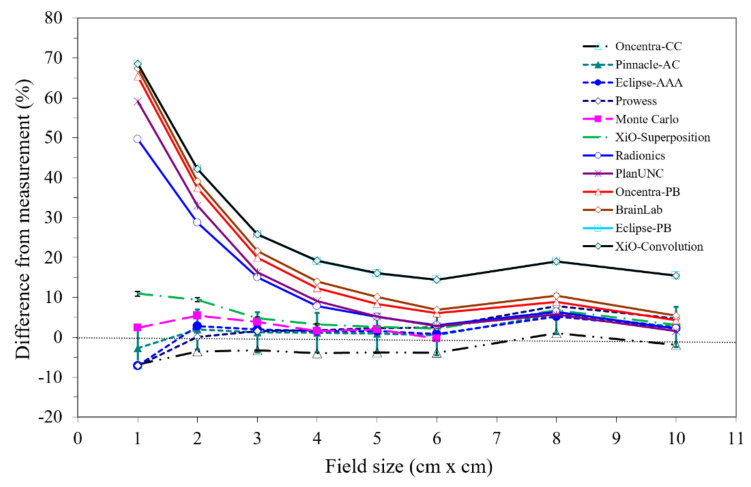
Differences between measured and TPS calculated doses from different algorithms at 10.5 cm depth. Graphs are separated in two groups by the type of algorithm. For a very small fields, very large differences (+70%) are noted in type-A (pencil beams) and smaller (±10%) in type-B algorithms (collapsed cone).

**Table 1 jcm-12-00906-t001:** Dose–volume constraints sent to institutions.

Structure	Constraints
PTV	95% dose 95% volume
	110% dose < 10% volume
	115% dose < 5% volume
Lung	V20 < 20%
Heart	50% dose < 70% volume
	65% dose < 2% volume
	75% dose < 0 volume
Spinal Cord	60% dose < 2% volume
	70% dose < 0% volume

**Table 2 jcm-12-00906-t002:** Details of the vendor, treatment-planning system, algorithm, and inhomogeneity correction algorithms for each treatment-planning system.

Vendor	TPS/Version	Algorithm	Treatment Machine
BrainLab	BrainScan/V5.31	Pencil beam convolution	Varian 2100EX
CMS	XiO/V4.3.1	Superposition/PBC	Varian Trilogy
Nomos *	Corvous V3	Clarkson Integration	Siemens Oncor
Varian	Eclipse V13.2	Pencil beam/AAA	Varian TrueBeam
MDS-Nordion *	Helax-TMS, V 6.0	Pencil Beam	Siemens Primus
Siemens KonRad *	KonRad V 2.2	Pencil Beam	Siemens Primus
Nucletron	Oncentra/V3.0	Pencil beam/CC	Siemens Oncor
Philips	Pinnacle V 9.7	Convolution/PBC	Elekta-Infinity
Nucletron *	Oncentra Plato V 13.7	Pencil Beam	Elekta-SL
Prowess *	Panther/V5.2	Convolution/PBC	Elekta-SL
Radionics	V RT4	Pencil beam (modified)	Varian 2100EX
Univ. of North Carolina	PLanUNC	Modified Batho, PB	Siemens Primus
Monte Carlo	Penelope/2002	Mixed MC Scheme	Varian Trilogy

* Not commonly used or obsolete.

## Data Availability

Data available on request.

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
