# Peer review of "Variability of Low-Z Inhomogeneity Correction in IMRT/SBRT: A Multi-Institutional Collaborative Study"

_jcm, 2023, doi:10.3390/jcm12030906_

Round 1
Reviewer 1 Report
I reviewed the manuscript " Variability of Low-Z Inhomogeneity Correction in IMRT/SBRT: A Multi-Institutional Collaborative Study" that analyzes calculated doses by different algorithms in a low-Z phantom, assessing criticalities and providing useful suggestions.
The topic is of interest for the purpose of the Journal.
The manuscript is well written, the presented methodology is generally rigorous with interesting results and discussion.
Despite I have some relatively minor criticisms that should be addressed, it was a pleasure to read this interesting and well-structured work.
Minor Criticisms:
1) Throughout the manuscript, I suggest to cite the “ICRU REPORT 91 Prescribing, Recording, and Reporting of Stereotactic Treatments with Small Photon Beams”, as example at lines 28, 65, 185, 204.
2) Materials and Methods, pag. 3, l. 98: the acronym IRB has to be given with the extended name as it cited for the first time in the text.
3) Materials and Methods, pag. 3, l. 98-198: please, explain the reason why the authors decided to use both a phantom and a CT dataset.
4) Materials and Methods, pag. 3, l. 101: please, insert a reference for the lung density and motivate better the choice of a material for a density of 0.25 g/cm3.
5) Materials and Methods, pag. 3, l. 104: please, provide a reference for the used dose-volume constrains. I suggest also to summarize these constrain in a table for sake of readability.
6) Materials and Methods, pag. 3, l. 116-118: the sentence “Jones et al ... density medium” does not refer to Materials and Methods section. Please move this part in the Introduction or in the Discussion section.
7) Materials and Methods, Table 1 and Figure 1: the names of Vendor/TPS reported in Table 1 and in the labels of Figure 1 have to be consistent. Some TPS, such as Corvus, Helax, Konrad, Plato reported in Figure 1 are missing in Table 1. Please correct Table 1 and Figure 1 in light of this observation and report the labels name of Figure 1 with the name of TPS, avoiding to use a mix of vendors and TPS names.
8) Results, pag. 4, l. 137-142: this sentence does not refer to Results section. Please move this part in the Materials and Methods section.
9) Results, pag. 5, l. 159: I think that the correct value is 1.65, not 16.5. Please correct.
10) It is difficult to review pag. 6, l. 172-185 as the Figure 4 is missing in the manuscript pdf file. Please provide Figure 4.
11) Results, pag. 7, ll. 190-198: these sentences are not understandable. Please, rephrase.
12) Results, Figure 5: same observation of point 7).
13) Discussion: the main limitation of this work lies in the usage of only one CT set instead of a cohort of patients CT data. Please discuss critically the limitation of this method.
Author Response
Comments and Suggestions for Authors
I reviewed the manuscript " Variability of Low-Z Inhomogeneity Correction in IMRT/SBRT: A Multi-Institutional Collaborative Study" that analyzes calculated doses by different algorithms in a low-Z phantom, assessing criticalities and providing useful suggestions.
The topic is of interest for the purpose of the Journal.
Response: Thanks for spending your valuable time during Holidays which is greatly appreciated. Also thanks for your comments.
The manuscript is well written, the presented methodology is generally rigorous with interesting results and discussion.
Response: Thanks.
Despite I have some relatively minor criticisms that should be addressed, it was a pleasure to read this interesting and well-structured work.
Response: Thanks for the compliments.
Minor Criticisms:
1) Throughout the manuscript, I suggest to cite the “ICRU REPORT 91 Prescribing, Recording, and Reporting of Stereotactic Treatments with Small Photon Beams”, as example at lines 28, 65, 185, 204.
Response: Thanks for your comment. We have now added ICRU 91 reference and provided statement as suggested whenever in the context of IMRT/VMAT dose reporting. As ICRU-91 does not specify algorithm, it is not suitable to refer every places as suggested.
2) Materials and Methods, pag. 3, l. 98: the acronym IRB has to be given with the extended name as it cited for the first time in the text.
Response: Text corrected.
3) Materials and Methods, pag. 3, l. 98-198: please, explain the reason why the authors decided to use both a phantom and a CT dataset.
Response: It is elaborated. Phantom was primarily used to measure the dose to compare with the TPS algorithm which is not possible in CT data of a patient.
4) Materials and Methods, pag. 3, l. 101: please, insert a reference for the lung density and motivate better the choice of a material for a density of 0.25 g/cm3.
Response: Corrected. It is commercially available and has density representative from 0.1-0.3 g/cm3 (human lung).
5) Materials and Methods, pag. 3, l. 104: please, provide a reference for the used dose-volume constrains. I suggest also to summarize these constrain in a table for sake of readability.
Response: This constraint was used by our institution. We have added a table as suggested.
6) Materials and Methods, pag. 3, l. 116-118: the sentence “Jones et al ... density medium” does not refer to Materials and Methods section. Please move this part in the Introduction or in the Discussion section.
Response: Thanks for the comment. Corrected.
7) Materials and Methods, Table 1 and Figure 1: the names of Vendor/TPS reported in Table 1 and in the labels of Figure 1 have to be consistent. Some TPS, such as Corvus, Helax, Konrad, Plato reported in Figure 1 are missing in Table 1. Please correct Table 1 and Figure 1 in light of this observation and report the labels name of Figure 1 with the name of TPS, avoiding to use a mix of vendors and TPS names.
Response: Thanks for picking this inconsistency. It is now corrected. Also table is updated.
8) Results, pag. 4, l. 137-142: this sentence does not refer to Results section. Please move this part in the Materials and Methods section.
Response: Corrected.
9) Results, pag. 5, l. 159: I think that the correct value is 1.65, not 16.5. Please correct.
Response: Corrected.
10) It is difficult to review pag. 6, l. 172-185 as the Figure 4 is missing in the manuscript pdf file. Please provide Figure 4.
Response: Corrected.
11) Results, pag. 7, ll. 190-198: these sentences are not understandable. Please, rephrase.
Response: As figure 4 as not visible a section of legend was left over this text. We have proof read the JCM formatted text in revised version of the text. Thanks.
12) Results, Figure 5: same observation of point 7).
Response: Corrected. Please note our comment on the measurements for justification.
13) Discussion: the main limitation of this work lies in the usage of only one CT set instead of a cohort of patients CT data. Please discuss critically the limitation of this method.
Response: Thank you for comments. This work is mainly to measure data related to algorithms which is not possible in patient. Patient data was included to show one case. We have now elaborated in the text.

Reviewer 2 Report
Modern radiotherapy techniques are increasing in small fields. This has driven the development of small field precision dose algorithms. This study showed that pencil beam algorithm does not provide an accurate dose and should not be considered with small fields. Using various algorithms, equipment, and treatment machine to verify.
But your study just mention that the type A algorithm is not suitable for TPS of lung cancer treatment, many studies are now proposing pencil beam modification algorithms, you may discuss or validate some modified algorithm in your study.
Author Response
Modern radiotherapy techniques are increasing in small fields. This has driven the development of small field precision dose algorithms. This study showed that pencil beam algorithm does not provide an accurate dose and should not be considered with small fields. Using various algorithms, equipment, and treatment machine to verify.
Response: Corrected.
But your study just mention that the type A algorithm is not suitable for TPS of lung cancer treatment, many studies are now proposing pencil beam modification algorithms, you may discuss or validate some modified algorithm in your study.
Response: Many of the inconsistencies have been edited and corrected.

Reviewer 3 Report
Overall, the article is not novel and lacks impact, as its main conclusion is a well-known fact that was already reported in the literature. Many reports by AAPM, IAEA, and RTOG already acknowledged this fact and make some recommendations.
line 125 and figure 1: Not sure if this could support the conclusion, as the authors stated there were multiple factors, such as the optimization methods, the optimization parameters, and how the planner make the plan, very likely, these plans were not the same even if they used the same cost function. So, I can't agree that the comparison among DVHs can support that the dose calculation algorithm plays a paramount role here.
line 159 correction factor 16.5 is this correct?
the author inconsistently uses inhomogeneity correction and the correction factor should keep this consistent throughout the article.
figure 4 is missing
Author Response
Overall, the article is not novel and lacks impact, as its main conclusion is a well-known fact that was already reported in the literature. Many reports by AAPM, IAEA, and RTOG already acknowledged this fact and make some recommendations.
Response: Let us greatly acknowledge your time end efforts during Holidays to review our work. We agree that it is not unique or novel and known to some readers, however, it is still being used. We provided ample of reason and measured data to reflect the scientific reason not to use pencil beam.
line 125 and figure 1: Not sure if this could support the conclusion, as the authors stated there were multiple factors, such as the optimization methods, the optimization parameters, and how the planner make the plan, very likely, these plans were not the same even if they used the same cost function. So, I can't agree that the comparison among DVHs can support that the dose calculation algorithm plays a paramount role here.
Response: You are right that is why we did phantom study and provided data to support.
line 159 correction factor 16.5 is this correct?
Response: Corrected. It should be 1.65
the author inconsistently uses inhomogeneity correction and the correction factor should keep this consistent throughout the article.
Response: Corrected in the text.
Figure 4 is missing
Response: Sorry about figure 4. It was simply lost during formatting. Now it is included.

Round 2
Reviewer 3 Report
Would the authors please give more details about the survey? (the number of hospitals, the distribution of geographical locations, academic vs privacy, how the survey is conducted, etc). I think readers may want to get an idea of how representative the sample is since the impact of the paper relies on the argument that pencil beam algorithms (26%) are currently widely used.
Author Response
We greatly appreciate your remarks. We have now provided the details of our survey in the method section.
We have also figure 1 for cork phantom as suggested by editor.